# Survey on Sheep Usage in Biomedical Research

**DOI:** 10.3390/ani10091528

**Published:** 2020-08-30

**Authors:** Corina Mihaela Berset, Urban Lanker, Stephan Zeiter

**Affiliations:** 1AO Research Institute Davos, Clavadelerstrasse 8, CH-7270 Davos, Switzerland; corina.berset@yahoo.com (C.M.B.); urban.lanker@aofoundation.org (U.L.); 2Animal Welfare and 3R Department, University of Zurich, Winterthurerstrasse 190, CH-8057 Zürich, Switzerland

**Keywords:** sheep, survey, preclinical research, health

## Abstract

**Simple Summary:**

Sheep are used in biomedical research. A European survey was conducted with the goal of identifying the need for improvement in the use of sheep. Most participants were veterinarians working at academic institutions. Two thirds have been working with sheep for more than 5 years, and their answers emphasized the importance of healthy sheep to be used in biomedical research, as about 60% have encountered health problems not related to study protocol. Other important points were sheep availability and the trust into and experience of the sheep supplier. This survey identified important points for refinement in the use of sheep in biomedical research with health status and monitoring as possible starting points.

**Abstract:**

Currently, there is a lack of detailed information about sheep used for biomedical research. Therefore, a European survey was conducted among sheep users gathering information on the current situation, with emphasis on animal selection criteria and issues encountered in practice. The ultimate goal was to identify needs for improvement, which will subsequently lead to a refinement and reduction of the total number of animals used for experimental studies. From the 84 respondents, 77.4% were veterinarians, 71.4% were employed at academic institutions and 63.1% had worked with sheep as research animals for more than 5 years. The majority of the respondents were using females (79.8%) with no clear age preference, mainly for surgical procedures and testing medical devices. The main criteria for choosing a sheep supplier were the animals’ health status, their availability, the trust and experience in the sheep provider and the animals’ uniformity. Approximately 60% of the respondents had encountered problems in their sheep not related to the experimental protocol and almost half of them did not have a health monitoring program for their animals. In conclusion, there is definitely a need for refinement in selecting sheep used in biomedical research, with their health status as possible starting point.

## 1. Introduction

According to European Union (EU) reports [1], approximately 20,000 sheep are enrolled in research projects each year in EU member countries. Article 10 of the EU Directive 2010/63 states that a member state shall ensure that inter alia laboratory rodents, rabbits, dogs and cats, but not sheep or other farm animals, may only be used in studies where those animals have been bred for use in studies [2]. Moreover, to the authors’ knowledge, there are currently no publications regarding the detailed criteria for selecting sheep for biomedical research. The health status of the animals is often only briefly described (i.e., clinically healthy) and there are no reports of compliance to the recommendations for the health monitoring of ruminants of the Federation of European Laboratory Animal Science Associations (FELASA) published in 2000 [3].

The goal of this European survey among sheep users was to gather information on the current situation, with emphasis on animal selection criteria and issues encountered in practice. Our motivation was to identify needs for improvement which will subsequently lead to a refinement and reduction of the total number of sheep used for experimental studies.

## 2. Materials and Methods

A survey was conducted between September 2016 and January 2017 among sheep users in biomedical research, with a focus on EU countries and Switzerland. The aim of the questionnaire was to gain more in-depth knowledge about the field of research in which sheep are used, the level of experience of the users, the animal characteristics and selection criteria as well as the issues encountered in practice.

The survey comprised 16 questions and was initially distributed as hard copies at the Swiss Laboratory Animal Science Association (SGV) Annual Meeting, Basel, Switzerland (13–14 September 2016). At the beginning of October 2016, an online version of the survey was created using surveymonkey.com in English (Appendix A) and in French (Appendix A) and distributed with the aid of several European Societies and networks: the Swiss Laboratory Animal Science Association (SGV), the Swiss Animal Welfare Officers Network, the European Society of Laboratory Animal Veterinarians (ESLAV), the European College of Laboratory Animal Medicine (ECLAM), the French Association for Laboratory Animal Science & Techniques (AFSTAL) and the Email for Vets in Laboratory Animal Medicine List (VOLE). A total of 84 responses were collected before the end of January 2017, when the survey was closed.

All answers were anonymous; the participants could write comments while answering some of the questions and at the end of the survey. After collecting all the answers, the data were exported to an Excel workbook.

## 3. Results

Among the respondents, 77.4% were veterinarians and 6% were animal care takers or technicians. From the total number of participants, 31% were conducting research activities and 15.5% were animal facility managers (multiple roles possible). Most of the respondents were working at academic institutions when the survey was conducted (71.4%), 15.5% were working at private companies, while 13.1% were working at other types of institutions.

The personnel’s experience, together with education and training, being a very important factor when performing in vivo research, the participants were asked about their previous experience working with sheep. According to their answers, at the time that they filled out the questionnaire, 63.1% had worked with sheep for at least 5 years, 28.6% for 1–5 years and 8.3% for less than 1 year. The majority of institutions or companies where they were employed (86.9%) had more than 5 years of experience in working with sheep, 8.3% had used sheep for 1–5 years and 4.8% for less than 1 year.

When asked about the number of animals used/year, 29.8% of the respondents reported between 1 and 20 animals, 26.2% were using between 21 and 50 animals, 14.3%, 51–100 animals and 27.4% more than 100 sheep/year.

In order to have a more precise overview of the sheep biomedical studies, the survey also aimed to identify the experimental field in which the animals were used; the answers are illustrated in Figure 1.

When questioned about the origin of the sheep they were using, the respondents reported purchasing the sheep either from local farms (59.5%), commercial breeders (19%) or had their own flock (21.5%).

As the sex bias in biomedical research is an important topic that has been frequently addressed lately [4], the sheep users were also asked about the sex of the animals they used for their studies. The majority were using females (79.8%), 19% used neutered males, 16.7% intact males, while, for 19%, either the sex of the animals was not important or animals were selected based on availability. For this question, multiple answers were possible and thus the percentages add up a total of 134%, indicating both female and male (both intact and neutered) were used by some institutions.

Regarding the age range of the sheep at the beginning of the study, 46.4% of the respondents were using animals that were 1–2 years old, 33.3% older than 2 years, 32.2% younger than 1 year old, while 13.1% did not consider this to be important or used the animals that were available. Multiple answers were possible, and, based on the total percentage (125%), sheep of different ages are used by some respondents. Approximately half of the respondents (51.2%) did not have any preference for the sheep breed.

The main criteria for choosing a sheep supplier has not been reported in biomedical publications so far. The respondents were asked to rate several criteria in the order of importance. The animals’ health status ranked first, followed by their availability, the trust and experience in the sheep provider and the animal homogeneity/uniformity (Figure 2).

Only 51.2% of the participants had a health monitoring program for their sheep. Among the diseases for which the animals had been screened, Q Fever was the most frequently cited. Half of the respondents reported that they were vaccinating their sheep (overall, almost one third had vaccinated against clostridia).

When asked if they had encountered problems in their sheep not related to the experimental protocol, 57.1% answered positively and, except one participant that reported problems at lambing, all the others mentioned health problems in the comments section for this question.

More than half of the respondents did not know the current purchase cost for their sheep; 67.9% could not estimate how much they would be willing to pay for a sheep with a controlled health status. The other 32.1% proposed prices ranging from EUR 50 to 1000.

## 4. Discussion

The goal of this survey among sheep users was to gather information on the current situation, with emphasis on animal selection criteria and issues encountered in practice if sheep are used for biomedical research. This survey has revealed that, currently, there is a strong need for improvement regarding sheep selection for biomedical research. The major issue identified was the animals’ health status, which was, unfortunately, often insufficiently known when the animals were enrolled in experiments. With about 60% of the participants reporting on having encountered health issues unrelated to the experimental protocol, this seems to be a common problem. This may have a great impact on the experimental results, on the reproducibility of the findings and engender unnecessary ethical and scientific costs.

In this survey, only half of the participants followed a health monitoring program in sheep, whereas such programs are considered to be state of the art in rodent facilities. Furthermore, if there are sheep monitoring program in place, they differ substantially from institution to institution and are focused on only a few pathogens. It has to pointed out that the EU Directive 2010/63 states Annex III (requirements for establishments and for the care and accommodation of animals) in Section 3.1 that “establishments shall have a strategy in place to ensure that a health status of the animals is maintained that safeguards animal welfare and meets scientific requirements. This strategy shall include regular health monitoring, a microbiological surveillance program and plans for dealing with health breakdowns and shall define health parameters and procedures for the introduction of new animals” [2]. Beside the legal requirement, in the authors’ opinion, health and welfare monitoring is refinement. The definition of “refinement” in animal experimentation has evolved significantly in recent decades. Russel and Burch, the creators of the 3Rs concept, described as refinement “simply to reduce to an absolute minimum the amount of stress imposed on those animals that are still used” [5]. They further characterized refinement as “an art or an ability to improvise”, mentioning that “the greatest experimenters have been artists in this sense”. Several decades later, Buchanan-Smith et al. [6] proposed a “harmonised progressive definition” of refinement, “in line with changes in animal ethics and animal welfare science” that included “health and welfare monitoring”. Unfortunately, in the 21st century, despite guidelines and recommendations from experts [3,7,8,9], there are still no reports of the implementation of this harmonized definition in terms of health monitoring for sheep used for research purposes. The recently published recommendations of best practices for the health management of ruminants and pigs used for scientific and educational purposes of the FELASAmight be a first step in establishing best health and welfare management practices (i.e., refinement) at institutions [10].

Considering that the animals’ availability ranked second among the criteria for selecting a sheep provider, it could be useful in the future to have specialized sheep breeders for biomedical research that ensure a good level of control and knowledge of the animals’ health status.

## 5. Conclusions

This survey identifies health status, health monitoring and sheep provider as starting points for refinement in the use of sheep for biomedical research. This is in line with the harmonized, progressive definition of refinement [6].

## Figures and Tables

**Figure 1 animals-10-01528-f001:**
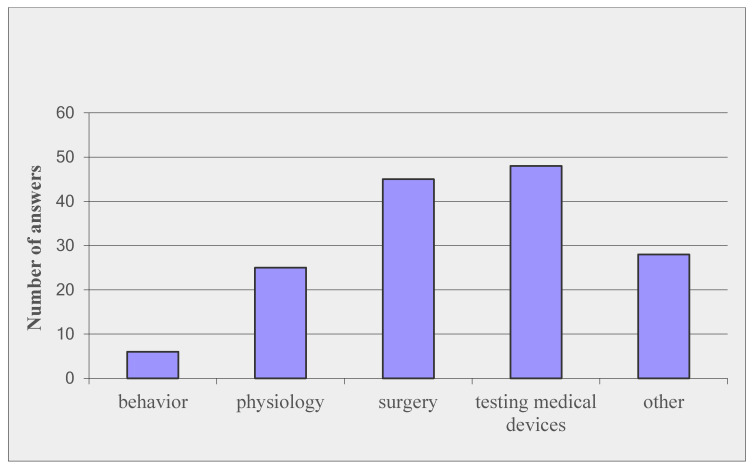
Main field of research using sheep (multiple answers possible).

**Figure 2 animals-10-01528-f002:**
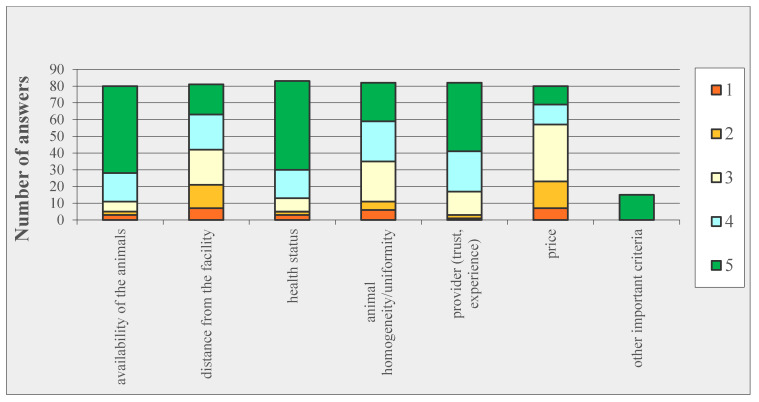
Main criteria for choosing a sheep supplier (several options were possible, rated from 5 (very important) to 1 (not important)).

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
