# Peer review of "Survey on Sheep Usage in Biomedical Research"

_animals, 2020, doi:10.3390/ani10091528_

Round 1
Reviewer 1 Report
Dear team, thank you for an interesting manuscript - and some fascinating and frightening findings! (Frightening because I work with sheep in my research and am very glad to say that we do have good health monitoring in place - unlike the majority!)
I only have a few questions for clarification:
Lines 69 and 70, can you describe the remaining 17.6 % (line 69) and 53.5% (line 70)?
Line 93 suggest: "either the sex of the animals was not important, or animals were selected based on availability"
Lines 98-101 and Figure 2. I was a little confused here - you open this section with "one very important aspect" (one being the important word), and then list four top-ranking criteria, and figure 2 shows 6... which one is very important... or are all 6 important but not routinely reported?
Line 107, delete the 'for' before Q-fever - you already have the correct grammar earlier with "for which the animals had been screened"
Once again, thank-you for this fascinating insight into the reality of animal experimentation.
Author Response
Dear reviewer
thank you for your constructive feedback and indeed these are frightening findings.
This is our point-by-point response to your comments:
Lines 69 and 70, can you describe the remaining 17.6 % (line 69) and 53.5% (line 70)?
As multiple answers were possible for the second question of the survery "What is your role at your instituion/ company?" these numbers do not add up to 100%. To clarify this we added in brackets (multiple roles possible) to the manuscript. 8% answered with "other", which we do not report separately.
Line 93 suggest: "either the sex of the animals was not important, or animals were selected based on availability"
We changed this sentence as suggested.
Lines 98-101 and Figure 2. I was a little confused here - you open this section with "one very important aspect" (one being the important word), and then list four top-ranking criteria, and figure 2 shows 6... which one is very important... or are all 6 important but not routinely reported?
With "one aspect" we meant the criteria for choosing a sheep supplier which has not been reported yet. We agree that this is confusing and changed the sentence to:"The main criteria for choosing a sheep supplier has not been reported in biomedical publications so far."
Line 107, delete the 'for' before Q-fever - you already have the correct grammar earlier with "for which the animals had been screened"
Thank you for pointing this out - we changed it accordingly.
We hope that our answers address your points appropriately and thank you once again for your constructive input.
Best regards
Stephan Zeiter
Reviewer 2 Report
Dear authors,
The article offers a study on the use of sheep for research, which is a novel topic that may be of interest to the scientific community. It is a short communication that provides little information in the introduction and discussion on other publications, possibly because there is almost nothing published on the use of sheep, but perhaps it could refer to what is done and applied in other species.
Regarding the results, several of the percentages shown are inconsistent. For example, "The majority were using females (79.8 %), 19% used neutered males, 16.7%: intact males, while for 19 %, either the sex of the animals was not important, either the animals that were available were used." I can understand that the 19% that don't mind the sex could be among the other percentages, but the other percentages add up to more than 100%, does it mean that some of the researchers are using females an males at the same time? This should be better explained in the text. It happens the same with the age of the used animals. Further, at the beginning of results, it is said that 77.4% were veterinarians and 6% were technicians, what happened with the others? what were they?
In line 80, the order in which these results are explained does not make much sense. Or they should be ordered by the number of sheep used (the most logical) or by the importance of the results.
In general, the results should be improved.
I find the article interesting, but it needs a major revision, improving results and adding some references to other related species.
Regards.
Author Response
Dear reviewer
thank you for your valuable input to our manuscript. Please find below a point-by-point answer to your comments:
Regarding the results, several of the percentages shown are inconsistent. For example, "The majority were using females (79.8 %), 19% used neutered males, 16.7%: intact males, while for 19 %, either the sex of the animals was not important, either the animals that were available were used." I can understand that the 19% that don't mind the sex could be among the other percentages, but the other percentages add up to more than 100%, does it mean that some of the researchers are using females an males at the same time? This should be better explained in the text. It happens the same with the age of the used animals.
For both answers multiple responses were possible- therefore the number do not necessarily have to sum up to 100%. We agree with the reviewer, that because the total sum is more than 100%, some researchers use male and female sheep at their institution. We added the following sentences to the manuscript:
"For this question, multiple answers were possible and because the percentages add up to a total of 134%, in some institutions both female and male (both intact and neutered) are used. "
"Multiple answers were possible and based on the total percentage (125%) sheep of different ages are used by some respondents."
Further, at the beginning of results, it is said that 77.4% were veterinarians and 6% were technicians, what happened with the others? what were they?
Unfortunately, this is again a question with multiple answers. To clarify we added "(multiple roles possible)". to the manuscript.
In line 80, the order in which these results are explained does not make much sense. Or they should be ordered by the number of sheep used (the most logical) or by the importance of the results.
We agree with this comment and changed to order by the number of sheep used.
In general, the results should be improved.
The authors hope that with the added clarifications according to the points raised the results section is now easier to understand. If there are sections that are difficult to understand, please indicate where they are.
The article offers a study on the use of sheep for research, which is a novel topic that may be of interest to the scientific community. It is a short communication that provides little information in the introduction and discussion on other publications, possibly because there is almost nothing published on the use of sheep, but perhaps it could refer to what is done and applied in other species.
On one hand, as the reviews points out, there is no such information available regarding the use of sheep in biomedical research and on the other hand, the situation in other laboratory animal species (i.e. mice and rats) is so distinct different, that in the authors`opinion one would compare apple with oranges. Furthermore, practices in those species could not be applied to sheep used for biomedical research due to practical (no commercial breeders at the moment) and financial reasons. However, the authors added to the manuscript that ensurance of the health status is one of requirements stated in the EU Directive 2010/63 and that the recently published recommendation of the FELASA working group could be the first step in establishing best health and welfare management practices at institutions.
"It has to pointed out that the EU Directive 2010/63 states Annex III (Requirements for establishments and for the care and accommodation of animals) in section 3.1 that "establishments shall have a strategy in place to ensure that a health status of the animals is maintained that safeguards animal welfare and meets scientific requirements. This strategy shall include regular health monitoring, a microbiological surveillance programme and plans for dealing with health breakdowns and shall define health parameters and procedures for the introduction of new animals"[4]."
"The recently published recommendations of best practices for the health management of ruminants and pigs used for scientific and educational purposes of the Federation of European Laboratory Animal Science Associations (FELASA) might be a first step in establishing best health and welfare management practices (i.e. refinement) at institutions [8]."
We believe that the suggested changes improved our short communication and we are looking forward to your answer.
Best regards
Stephan Zeiter
Round 2
Reviewer 2 Report
Dear authors,
Thank you very much for taking into account my suggestions. In my opinion, the results are better explained now.
Regarding discussion and the references to other species, I was not thinking of rats or mice, that, effectively, are too far from sheep, I was thinking, for example, of cattle and pigs. I have done a fast check and I found some references related to these species that could be added. This below could serve as an example:
Research with Agricultural Animals and Wildlife Por: Cox, Rebecca J.; Nol, Pauline; Ellis, Christine K.; et ál.. ILAR Journal 2020, vol 60, No, 1, 66-7Best regards.